# PLAN ONLINE, LEARN OFFLINE: EFFICIENT LEARNING AND EXPLORATION VIA MODEL-BASED CONTROL

**Kendall Lowrey**[*1]     **Aravind Rajeswaran**[*1]

**Sham Kakade**[1]     **Emanuel Todorov**[1,2]     **Igor Mordatch**[3]

[*] Equal contributions    [1] University of Washington    [2] Roboti LLC    [3] OpenAI

`klowrey, aravraj, sham, todorov @cs.uw.edu, mordatch@openai.com`

## ABSTRACT

We propose a "plan online and learn offline" framework for the setting where an agent, with an internal model, needs to continually act and learn in the world. Our work builds on the synergistic relationship between local model-based control, global value function learning, and exploration. We study how local trajectory optimization can cope with approximation errors in the value function, and can stabilize and accelerate value function learning. Conversely, we also study how approximate value functions can help reduce the planning horizon and allow for better policies beyond local solutions. Finally, we also demonstrate how trajectory optimization can be used to perform temporally coordinated exploration in conjunction with estimating uncertainty in value function approximation. This exploration is critical for fast and stable learning of the value function. Combining these components enable solutions to complex control tasks, like humanoid locomotion and dexterous in-hand manipulation, in the equivalent of a few minutes of experience in the real world.

## 1    INTRODUCTION

We consider a setting where an agent with limited memory and computational resources is dropped into a world. The agent has to simultaneously act in the world and learn to become proficient in the tasks it encounters. Let us further consider a setting where the agent has some prior knowledge about the world in the form of a nominal dynamics model. However, the state space of the world could be very large and complex, and the set of possible tasks very diverse. This complexity and diversity, combined with limited computational capability, rules out the possibility of an *omniscient* agent that has experienced all situations and knows how to act optimally in all states, even if the agent knows the dynamics. Thus, the agent has to act in the world while learning to become competent.

Based on the knowledge of dynamics and its computational resources, the agent is imbued with a local search procedure in the form of trajectory optimization. While the agent would certainly benefit from the most powerful of trajectory optimization algorithms, it is plausible that very complex procedures are still insufficient or inadmissible due to the complexity or inherent unpredictability of the environment. Limited computational resources may also prevent these powerful methods from real-time operation. While the trajectory optimizer may be insufficient by itself, we show that it provides a powerful vehicle for the agent to explore and learn about the world.

Due to the limited capabilities of the agent, a natural expectation is for the agent to be moderately competent for new tasks that occur infrequently and skillful in situations that it encounters repeatedly by learning from experience. Based on this intuition, we propose the *plan online and learn offline (POLO)* framework for continual acting and learning. POLO is based on the tight synergistic coupling between local trajectory optimization, global value function learning, and exploration.

We will first provide intuitions for why there may be substantial performance degradation when acting *greedily* using an approximate value function. We also show that value function learning can be accelerated and stabilized by utilizing trajectory optimization integrally in the learning process, and that a trajectory optimization procedure in conjunction with an approximate value function can compute near optimal actions. In addition, exploration is critical to propagate global information in value function learning, and for trajectory optimization to escape local solutions and saddle

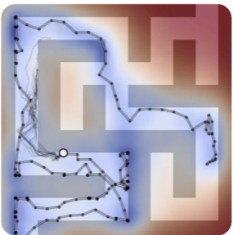 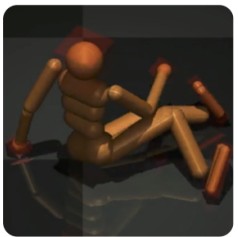 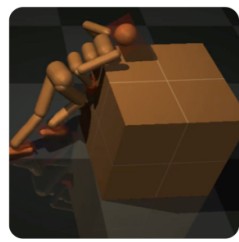 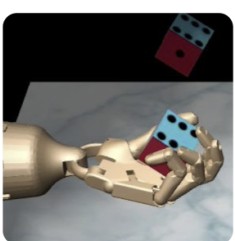

Figure 1: Examples of tasks solved with POLO. A 2D point agent navigating a maze without any directed reward signal, a complex 3D humanoid standing up from the floor, pushing a box, and in-hand re-positioning of a cube to various orientations with a five-fingered hand. Video demonstration of our results can be found at: https://sites.google.com/view/polo-mpc.

points. In POLO, the agent forms hypotheses on potential reward regions, and executes temporally coordinated action sequences through trajectory optimization. This is in contrast to strategies like $\epsilon-$greedy and Boltzmann exploration that explore at the granularity of individual timesteps. The use of trajectory optimization enables the agent to perform directed and efficient exploration, which in turn helps to find better global solutions.

The setting studied in the paper models many problems of interest in robotics and artificial intelligence. Local trajectory optimization becomes readily feasible when a nominal model and computational resources are available to an agent, and can accelerate learning of novel task instances. In this work, we study the case where the internal nominal dynamics model used by the agent is accurate. Nominal dynamics models based on knowledge of physics (Todorov et al., 2012), or through learning (Ljung, 1987), complements a growing body of work on successful simulation to reality transfer and system identification (Ross & Bagnell, 2012; Rajeswaran et al., 2016; Lowrey et al., 2018; OpenAI, 2018). Combining the benefits of local trajectory optimization for fast improvement with generalization enabled by learning is critical for robotic agents that live in our physical world to continually learn and acquire a large repertoire of skills.

## 2 THE POLO FRAMEWORK

The POLO framework combines three components: local trajectory optimization, global value function approximation, and an uncertainty and reward aware exploration strategy. We first present the motivation for each component, followed by the full POLO procedure.

### 2.1 DEFINITIONS, NOTATIONS, AND SETTING

We model the world as an infinite horizon discounted Markov Decision Process (MDP), which is characterized by the tuple: $\mathcal{M} = \{\mathcal{S}, \mathcal{A}, \mathcal{R}, \mathcal{T}, \gamma\}$. $\mathcal{S} \in \mathbb{R}^n$ and $\mathcal{A} \in \mathbb{R}^m$ represent the continuous (real-valued) state and action spaces respectively. $\mathcal{R} : \mathcal{S} \times \mathcal{A} \to \mathbb{R}$ represents the reward function. $\mathcal{T} : \mathcal{S} \times \mathcal{A} \times \mathcal{S} \to \mathbb{R}_+$ represents the dynamics model, which in general could be stochastic, and $\gamma \in [0, 1)$ is the discount factor. A policy $\pi : \mathcal{S} \times \mathcal{A} \to \mathbb{R}_+$ describes a mapping from states to actions. The value of a policy at a state is the average discounted reward accumulated by following the policy from the state: $V^\pi(s) = \mathbb{E}[\sum_{t=0}^{\infty} \gamma^t r(s_t, \pi(s_t)) \mid s_0 = s]$. The overall performance of the policy over some start state distribution $\beta$ is given by: $J^\beta(\pi) = \mathbb{E}_{s \sim \beta}[V^\pi(s)]$. For notational simplicity, we use $s'$ to denote the next state visited after (from) $s$.

As described earlier, we consider the setting where an agent is dropped into a complex world. The agent has access to an internal model of the world. However, the world can be complex and diverse, ruling out the possibility of an omniscient agent. To improve its behavior, the agent has to explore and understand relevant parts of the state space while it continues to act in the world. Due to the availability of the internal model, the agent can revisit states it experienced in the world and reason about alternate potential actions and their consequences to learn more efficiently.

## 2.2 VALUE FUNCTION APPROXIMATION

The optimal value function describes the long term discounted reward the agent receives under the optimal policy. Defining the Bellman operator at state $s$ as:

$$\mathcal{B}V(s) = \max_a \mathbb{E}\left[r(s, a) + \gamma V(s')\right], \tag{1}$$

the optimal value function $V^*$ corresponds to the fixed point: $V^*(s) = \mathcal{B}V^*(s) \; \forall s \in \mathcal{S}$. For small, tabular MDPs, classical dynamic programming algorithms like value iteration can be used to obtain the optimal value function. The optimal policy can be recovered from the value function as: $\pi^*(s) = \arg\max_a \mathbb{E}[r(s, a) + \gamma V^*(s')]$. For more complex MDPs, computing the optimal value function exactly is not tractable except in a few well known cases like the LQR (Åström & Murray, 2004) and LMDPs (Todorov, 2006; Dvijotham & Todorov, 2011). Thus, various approximate techniques have been considered in prior works. One popular approach is fitted value iteration (Bertsekas & Tsitsiklis, 1996; Munos & Szepesvári, 2008), where a function approximator (e.g. neural network) is used to approximate the optimal value function. The core structure of fitted value iteration considers a collection of states (or a sampling distribution $\nu$), and a parametric value function approximator $\hat{V}_\theta$. Inspired by value iteration, fitted value iteration updates parameters as:

$$\theta_{i+1} = \arg\min_\theta \; \mathbb{E}_{s \sim \nu}\left[\left(\hat{V}_\theta(s) - \mathcal{B}\hat{V}_{\theta_i}(s)\right)^2\right] \tag{2}$$

where $\mathcal{B}\hat{V}_{\theta_i}(s)$ are targets for the regression problem computed at the specific state $s$ according to Eq. (1). After sufficient iterations of the procedure in Eq. (2) to get a good approximation, the policy is recovered as $\hat{\pi}(s) = \arg\max_a \mathbb{E}[r(s, a) + \gamma\hat{V}_\theta(s')]$. The success and convergence of this overall procedure depends critically on at least two components: the capacity and structure of the function approximator ($\theta$); and the sampling distribution ($\nu$).

**Lemma 1.** *(Bertsekas & Tsitsiklis, 1996) Let $\hat{V}$ be an approximate value function with $\ell_\infty$ error $\epsilon := \max_s |\hat{V}(s) - V^*(s)|$. Let $\hat{\pi}(s) = \arg\max_a \mathbb{E}[r(s, a) + \gamma\hat{V}(s')]$ be the induced greedy policy. For all MDPs and $\beta$, the bound in Eq. (3) holds. Furthermore, for any size of the state space, there exist MDPs and $\hat{V}$ for which the bound is tight (holds with equality).*

$$J^\beta(\pi^*) - J^\beta(\hat{\pi}) \le \frac{2\gamma\epsilon}{1 - \gamma} \tag{3}$$

Intuitively, this suggests that performance of $\hat{\pi}$ degrades with a dependence on effective problem horizon determined by $\gamma$. This can be understood as the policy paying a price of $\epsilon$ at every timestep. Due to the use of function approximation, errors may be inevitable. In practice, we are often interested in temporally extended tasks where $\gamma \approx 1$, and hence this possibility is concerning. Furthermore, the $\arg\max$ operation in $\hat{\pi}$ could inadvertently exploit approximation errors to produce a poor policy. The performance of fitted value iteration based methods also rely critically on the sampling distribution to propagate global information (Munos & Szepesvári, 2008), especially in sparse reward settings. For some applications, it may be possible to specify good sampling distributions using apriori knowledge of where the optimal policy should visit (e.g. based on demonstration data). However, automatically generating such sampling distributions when faced with a new task may be difficult, and is analogous to the problem of exploration.

## 2.3 TRAJECTORY OPTIMIZATION AND MODEL PREDICTIVE CONTROL

Trajectory optimization and model predictive control (MPC) have a long history in robotics and control systems (Garcia et al., 1989; Tassa et al., 2014)[1]. In MPC, starting from state $s_t$ and using the knowledge of the dynamics model, a locally optimal sequence of actions (or policies) up to a moving horizon of $H$ is computed by solving the following optimization problem.

---

[1]In this work, we use the terms trajectory optimization and MPC interchangeably

$$\begin{aligned}
\underset{\{\tilde{\pi}_k\}_{k=t}^{t+H}}{\text{maximize}} \quad & \mathbb{E}\left[\sum_{k=t}^{t+H-1} \gamma^{(k-t)} r(\boldsymbol{x}_t, \boldsymbol{u}_t) + \gamma^H r_f(\boldsymbol{x}_{t+H})\right] \\
\text{subject to} \quad & \boldsymbol{x}_{k+1} \sim \mathcal{T}(\boldsymbol{x}_k, \boldsymbol{u}_k) \\
& \boldsymbol{u}_k \sim \tilde{\pi}_t(\cdot|\boldsymbol{x}_k) \\
& \boldsymbol{x}_t = s_t.
\end{aligned} \tag{4}$$

Here, we use $\boldsymbol{x}, \boldsymbol{u}, \tilde{\pi}$ as dummy variables for states, actions, and policy to distinguish the "imagined" evolution of the MDP used for the trajectory optimization with the actual states ($s$) observed in the true evolution of the MDP. Here, $r(\boldsymbol{x}, \boldsymbol{u})$ represents the running reward which is the same as the MDP reward function, and $r_f(\boldsymbol{x}_{t+H})$ represents a terminal reward function. Let $\{\tilde{\pi}_k^*\}$ be the local time-indexed policies obtained as the solution to the optimization problem in (4). After solving the optimization problem, the first local time-indexed policy is used as $\hat{\pi}_{MPC}(\cdot|s_t) := \tilde{\pi}_t^*(\cdot|\boldsymbol{x}_t)$. The entire procedure is repeated again in the next time step $(t+1)$. Note that we have defined the optimization problem over a sequence of feedback policies. However, if the dynamics is deterministic, a sequence of actions $\{\boldsymbol{u}_k\}_{k=t}^{t+H}$ can be optimized and used instead without any loss in performance. See Appendix C for further discussions. This approach has led to tremendous success in a variety of control systems such as power grids, chemical process control (Qina & Badgwellb, 2003), and more recently in robotics (Williams et al., 2016). Since MPC looks forward only $H$ steps, it is ultimately a local method unless coupled with a value function that propagates global information. In addition, we also provide intuitions for why MPC may help accelerate the learning of value functions. This synergistic effect between MPC and global value function forms a primary motivation for POLO.

**Impact of approximation errors in the value function**

**Lemma 2.** *Let $\hat{V}$ be an approximate value function with $\ell_\infty$ error $\epsilon := \max_s |\hat{V}(s) - V^*(s)|$. Suppose the terminal reward in Eq. (4) is chosen as $r_f(s_H) = \hat{V}(s_H)$, and let the MPC policy be $\hat{\pi}_{MPC}(\cdot|s_t) := \tilde{\pi}_t^*(\cdot|\boldsymbol{x}_t)$ (from Eq. 4). Then, for all MDPs and $\beta$, the performance of the MPC policy can be bounded as:*

$$J^\beta(\pi^*) - J^\beta(\hat{\pi}_{MPC}) \le \frac{2\gamma^H \epsilon}{1 - \gamma^H}. \tag{5}$$

*Proof.* The proof is provided in Appendix C. $\qquad\square$

This suggests that MPC (with $H > 1$) is less susceptible to approximation errors than greedy action selection. Also, without a terminal value function, we have $\epsilon = \mathcal{O}(r_{\max}/(1-\gamma))$ in the worst case, which adds an undesirable scaling with the problem horizon.

**Accelerating convergence of the value function**   Furthermore, MPC can also enable faster convergence of the value function approximation. To motivate this, consider the H-step Bellman operator: $\mathcal{B}^H V(s) := \max_{a_{0:H-1}} \mathbb{E}[\sum_{t=0}^{H-1} \gamma^t r_t + \gamma^H V(s_H)]$. In the tabular setting, for any $V_1$ and $V_2$, it is easy to verify that $|\mathcal{B}^H V_1 - \mathcal{B}^H V_2|_\infty \le \gamma^H |V_1 - V_2|_\infty$. Intuitively, $\mathcal{B}^H$ allows for propagation of global information for $H$ steps, thereby accelerating the convergence due to faster mixing. Note that one way to realize $\mathcal{B}^H$ is to simply apply $\mathcal{B}$ $H$ times, with each step providing a contraction by $\gamma$. In the general setting, it is unknown if there exists alternate, cheaper ways to realize $\mathcal{B}^H$. However, for problems in continuous control, MPC based on local dynamic programming methods (Jacobson & Mayne, 1970; Todorov & Li, 2005) provide an efficient way to *approximately* realize $\mathcal{B}^H$, which can be used to accelerate and stabilize value function learning.

## 2.4   PLANNING TO EXPLORE

The ability of an agent to explore the relevant parts of the state space is critical for the convergence of many RL algorithms. Typical exploration strategies like $\epsilon$-greedy and Boltzmann take exploratory actions with some probability on a *per time-step basis.* Instead, by using MPC, the agent can explore in the space of trajectories. The agent can consider a hypothesis of potential reward regions in the state space, and then execute the optimal trajectory conditioned on this belief, resulting in a

---

**Algorithm 1** Plan Online and Learn Offline (POLO)

---

1: **Inputs:** planning horizon $H$, value function parameters $\theta_1, \theta_2, \ldots \theta_K$, mini-batch size $n$, number of gradient steps $G$, update frequency $Z$
2: **for** $t = 1$ **to** $\infty$ **do**
3:     Select action $a_t$ according to MPC (Eq. 4) with terminal reward $r_f(s) \equiv \hat{V}(s)$ from Eq. (7)
4:     Add the state experience $s_t$ to replay buffer $\mathcal{D}$
5:     **if** $\mathrm{mod}(t, Z) = 0$ **then**
6:         **for** $G$ times **do**
7:             Sample $n$ states from the replay buffer, and compute targets using Eq. (8)
8:             Update the value functions using Eq. (6) (see Section 2.5 for details)
9:         **end for**
10:     **end if**
11: **end for**

---

temporally coordinated sequence of actions. By executing such coordinated actions, the agent can cover the state space more rapidly and intentionally, and avoid back and forth wandering that can slow down the learning. We demonstrate this effect empirically in Section 3.1.

To generate the hypothesis of potentially rewarding regions, we take a Bayesian view and approximately track a posterior over value functions. Consider a motivating setting of regression, where we have a parametric function approximator $f_\theta$ with prior $\mathbb{P}(\theta)$. The dataset consists of input-output pairs: $\mathcal{D} = (x_i, y_i)_{i=1}^n$, and we wish to approximate $\mathbb{P}(\theta|\mathcal{D})$. In the Bayesian linear regression setting with Gaussian prior and noise models, the solution to the following problem generates samples from the posterior (Osband et al., 2016; Azizzadenesheli et al., 2018a; Osband et al., 2018):

$$\arg\min_\theta ||\tilde{y}_i - f_{\tilde{\theta}}(x_i) - f_\theta(x_i)||_2^2 + \frac{\sigma^2}{\lambda}||\theta||_2^2 \qquad (6)$$

where $\tilde{y}_i \sim \mathcal{N}(y_i, \sigma^2)$ is a noisy version of the target and $\tilde{\theta} \sim \mathbb{P}(\theta)$ is a sample from the prior. Based on this, Osband et al. (2018) demonstrate the benefits of uncertainty estimation for exploration. Similarly, we use this procedure to obtain samples from the posterior for value function approximation, and utilize them for temporally coordinated action selection using MPC. We consider $K$ value function approximators $\hat{V}_\theta$ with parameters $\theta_1, \theta_2, \ldots \theta_K$ independently trained based on Eq. (6). We consider the softmax of the different samples as the value at a state:

$$\hat{V}(s) = \sum_{k=1}^K \omega_k(s)\hat{V}_{\theta_k}(s), \quad \text{where } \omega_k(s) \overset{\text{def}}{:=} \frac{\exp\left(\kappa\hat{V}_{\theta_k}(s)\right)}{\sum_{j=1}^K \exp\left(\kappa\hat{V}_{\theta_j}(s)\right)} \qquad (7)$$

Since the above scheme approximates mean + variance for small $\kappa > 0$, this procedure encourages the agent to additionally explore parts of the state space where the disagreement between the function approximators is large. This corresponds to the broad notion of optimism in the face of uncertainty (Auer et al., 2002) which has been successful in a number of applications (Silver et al., 2016; Li et al., 2010).

## 2.5 FINAL ALGORITHM

To summarize, POLO utilizes a global value function approximation scheme, a local trajectory optimization subroutine, and an optimistic exploration scheme. POLO operates as follows: when acting in the world, the agent uses the internal model and always picks the optimal action suggested by MPC. Exploration is implicitly handled by tracking the value function uncertainties and the optimistic evaluation, as specified in Eq. (6) and (7). All the experience (visited states) from the world are stored into a replay buffer $\mathcal{D}$, with old experiences discarded if the buffer becomes full. After every $Z$ steps of acting in the world and collecting experience, the value functions are updated by: (a) constructing the targets according to Eq. (8); (b) performing regression using the randomized prior scheme using Eq. (6) where $f_\theta$ corresponds to the value function approximator. For state $s$ in the buffer and value network $k$ with parameters $\theta_k$, the targets are constructed as:

$$y^k(s) = \max_{\{\tilde{\pi}_t\}_{t=0}^{N-1}} \mathbb{E}\left[\sum_{t=0}^{N-1} \gamma^t r(\boldsymbol{x}_t, \boldsymbol{u}_t) + \gamma^N \hat{V}_{\theta_k}(\boldsymbol{x}_N)\right], \quad \text{where } \boldsymbol{x}_0 = s, \boldsymbol{u}_t \sim \tilde{\pi}_t(\cdot|\boldsymbol{x}_t) \qquad (8)$$

which corresponds to solving a $N-$step trajectory optimization problem starting from state $s$. As described earlier, using trajectory optimization to generate the targets for fitting the value approximation accelerates the convergence and makes the learning more stable, as verified experimentally in Section 3.3. The overall procedure is summarized in Algorithm 1.

## 3 EMPIRICAL RESULTS AND DISCUSSION

Through empirical evaluation, we wish to answer the following questions:

1. Does trajectory optimization in conjunction with uncertainty estimation in value function approximation result in temporally coordinated exploration strategies?
2. Can the use of an approximate value function help reduce the planning horizon for MPC?
3. Does trajectory optimization enable faster and more stable value function learning?

Before answering the questions in detail, we first point out that POLO can scale up to complex high-dimensional agents like 3D humanoid and dexterous anthropomorphic hand (OpenAI, 2018; Rajeswaran et al., 2018) which are among the most complex control tasks studied in robot learning. Video demonstration can be found at: https://sites.google.com/view/polo-mpc

### 3.1 TRAJECTORY OPTIMIZATION FOR EXPLORATION

Exploration is critical in tasks where immediate rewards are not well aligned with long-term objectives. As a representative problem, we consider a point mass agent in different 2D worlds illustrated in figure 2: a simple finite size box with no obstacles and a maze. This domain serves to provide an intuitive understanding of the interaction between trajectory optimization and exploration while also enabling visualization of results. In the extreme case of no rewards in the world, an agent with only local information would need to continuously explore. We wish to understand how POLO, with its ensemble of value functions tracking uncertainties, uses MPC to perform temporally coordinated actions. Our baseline is an agent that employs random exploration on a per-time-step basis; MPC without a value function would not move due to lack of local extrinsic rewards. Second, we consider an agent that performs uncertainty estimation similar to POLO but selects actions greedily (i.e. POLO with a planning horizon of 1). Finally, we consider the POLO agent which tracks value uncertainties and selects actions using a 32-step MPC procedure. We observe that POLO achieves more region coverage in both point mass worlds compared to alternatives, as quantitatively illustrated in figure 2(a). The ensemble value function in POLO allows the agent to recognize the true, low value of visited states, while preserving an optimistic value elsewhere. Temporally coordinated action is necessary in the maze world; POLO is able to navigate down all corridors.

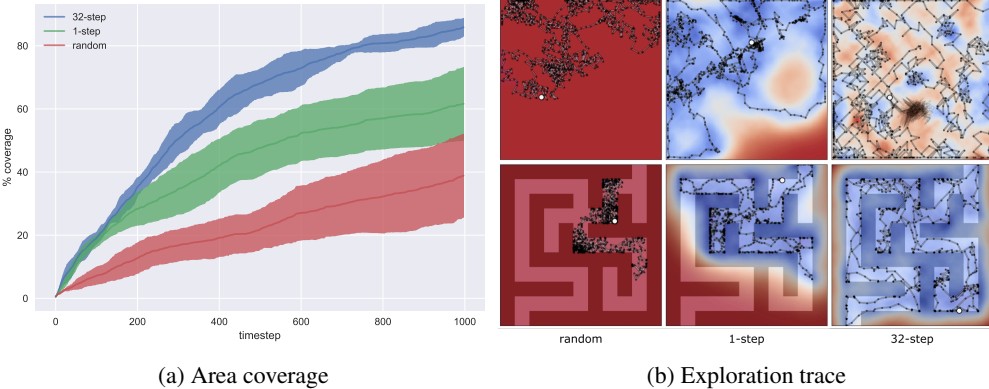

(a) Area coverage                    (b) Exploration trace

Figure 2: 2D point mass navigation task in a world with *no rewards*. Fig. (a) describes the percentage of an occupancy grid covered by the agent, averaged over 10 random seeds. Fig. (b) depicts an agent over 1000 timesteps; red indicates regions of high value (uncertainty) while blue denotes low. The value function learns to assign the true, low values to regions visited and preserves high values to unexplored regions; uncertainty and long horizons are observed to be critical for exploration.

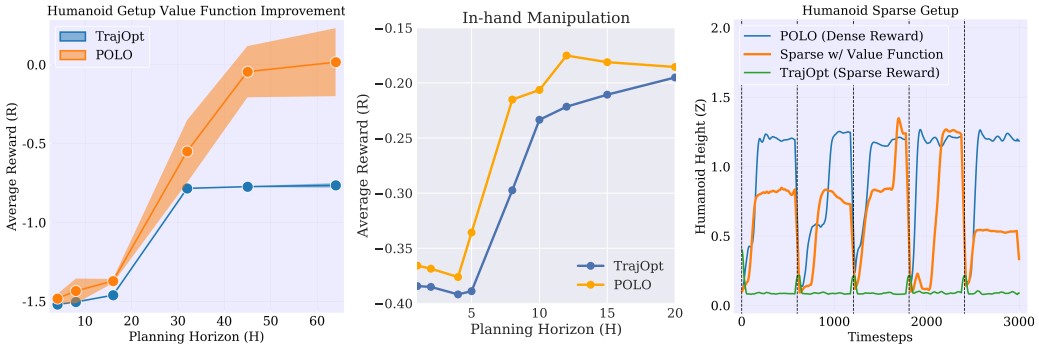

Figure 3: Performance as a function of planning horizon for the humanoid getup (left), and in-hand manipulation task (middle). POLO was trained for 12000 and 2500 environment timesteps, respectively. We test POLO with the learned terminal value function against pure MPC and compare average reward obtained over 3 trials in the getup task and 1000 steps in the manipulation task. On the right, a value function trained with POLO is used by MPC without per-time-step rewards. The agent's height increases, indicating a task-relevant value function. For comparison, we also include the trace of POLO with dense rewards and multiple trials (dashed vertical lines)

## 3.2 VALUE FUNCTION APPROXIMATION FOR TRAJECTORY OPTIMIZATION

Next, we study if value learning helps to reduce the planning horizon for MPC. To this end, we consider two high dimensional tasks: *humanoid getup* where a 3D humanoid needs to learn to stand up from the ground, and *in-hand manipulation* where a five-fingered hand needs to re-orient a cube to a desired configuration that is randomized every 75 timesteps. For simplicity, we use the MPPI algorithm (Williams et al., 2016) for trajectory optimization. In Figure 3, we consider MPC and the full POLO algorithm of the same horizon, and compare their performance after $T$ steps of learning in the world. We find that POLO uniformly dominates MPC, indicating that the agent is consolidating experience from the world into the value function. With even the longest planning horizon, the humanoid getup task has a local solution where it can quickly sit up, but cannot discover a chain of actions required to stand upright. POLO's exploration allows the agent to escape the local solution, and consolidate the experiences to consistently stand up. To further test if the learned value function is task aligned, we take the value function trained with POLO, and use it with MPC *without any intermediate rewards*. Thus, the MPC is optimizing a trajectory of length $H = 64$ purely using the value function of the state after 64 steps. We observe, in Figure 3, that even in this case, the humanoid is able to consistently increase its height from the floor indicating that the value function has captured task relevant details. We note that a greedy optimization procedure with this value function does not yield good results, indicating that the learned value function is only approximate and not good everywhere.

While the humanoid getup task presents temporal complexity requiring a large planning horizon, the in-hand manipulation task presents spatial complexity. A large number of time steps are not needed to manipulate the object, and a strong signal about progress is readily received. However, since the targets can change rapidly, the variance in gradient estimates can be very high for function approximation methods (Ghosh et al., 2018). Trajectory optimization is particularly well suited for such types of problems, since it can efficiently compute near-optimal actions conditioned on the instance, facilitating function approximation. Note that the trajectory optimizer is unaware that the targets can change, and attempts to optimize a trajectory for a fixed instance of the task. The value function consolidates experience over multiple target changes, and learns to give high values to states that are not just immediately good but provide a large space of affordances for the possible upcoming tasks.

## 3.3 TRAJECTORY OPTIMIZATION FOR VALUE FUNCTION LEARNING

Finally, we study if trajectory optimization can aid in accelerating and stabilizing value function learning. To do so, we again consider the humanoid getup task and study different variants of POLO. In particular, we vary the horizon $(N)$ used for computing the value function targets in Eq. (8). We

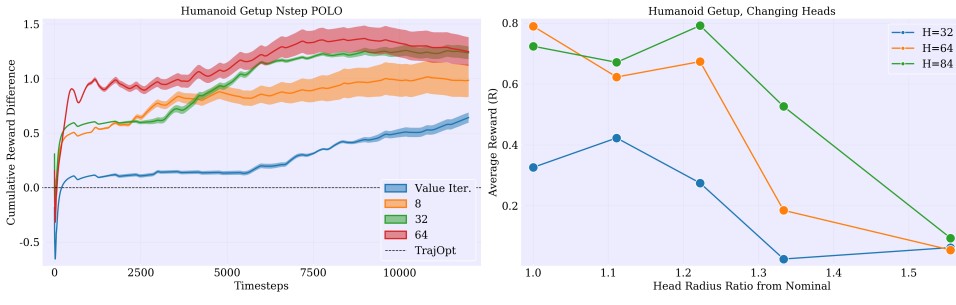

(a) POLO learning for different $N$-step horizons  (b) MPC with imperfect value function

Figure 4: Usefulness of trajectory optimization for value function learning. (a) illustrates that $N$-step trajectory optimization accelerates the learning of the value function. $N$=1 corresponds to trajectory centric fitted value iteration. A difference of $0.2$ reward to MPC amounts to approximately $50\%$ performance improvement. (b) value function trained for the nominal model (head size of $1.0$) used with MPC for models with larger sizes.

observe that as we increase $N$, the agent learns the value function with fewer interactions with the world, as indicated in Figure 4(a). The benefit of using $N-$step returns for stable value function learning and actor-critic methods have been observed in numerous works (Mnih et al., 2016; Munos et al., 2016; Schulman et al., 2016), and our experiments reinforce these observations. The use of $N-$step returns help to traverse the bias-variance trade-off. Furthermore, due to the discounting, the contribution of $V(s_N)$ is made weaker and thus the targets are more stable. This mirrors ideas such as target networks (Mnih et al., 2015) commonly used to stabilize training. As discussed earlier, longer horizons make trajectory optimization more tolerant to errors in the value function. To illustrate this, we take the value function trained with POLO on a nominal humanoid model, and perturb the model by changing the size of the head to model value function degradation. Figure 4(b) shows that a longer planning horizon can mitigate this degradation. This presents intriguing future possibility of using MPC to improve transfer learning between tasks or robot platforms.

# 4 RELATED WORK

**Planning and learning:** Combining elements of planning and search with approximate value functions has been explored in discrete game domains (Silver et al., 2017; Anthony et al., 2017) where an MCTS planner is informed by the value function. Alternatively, using prior data to guide the search process in continuous MCTS without explicitly learning a value function has also been explored (Rajamäki & Hämäläinen, 2017). Related to this, Atkeson (1993) uses an offline trajectory library for action selection in real-time, but do not explicitly consider learning parametric value functions. RTDP (Barto et al., 1995) considers learning value functions based on states visited by the agent, but does not explicitly employ the use of planning. Zhong et al. (2013) consider the setting of learning a value function to help MPC, and found the contribution of value functions to be weak for the relatively simple tasks considered in their work. Approaches such as cost shaping (Ng et al., 1999) can also be interpreted as hand specifying an approximate value function, and has been successfully employed with MPC (Tassa et al., 2012). However, this often require careful human design and task specific expertise. An alternative set of approaches (Ross et al., 2011; Levine & Koltun, 2013; Mordatch & Todorov, 2014; Sun et al., 2018b) use local trajectory optimization to generate a dataset for training a global policy through imitation learning. These approaches do not use MPC at runtime, and hence may often require retraining for changes in tasks or environment. Furthermore, results from this line of work have been demonstrated primarily in settings where trajectory optimization alone can solve the task, or use human demonstration data. In contrast, through our exploration schemes, we are able to improve over the capabilities of MPC and solve tasks where MPC is unsuccessful.

**Planning and exploration:** Exploration is a well-studied and important problem in RL. The importance of having a wide and relevant state distribution has been pointed out in numerous prior works (Munos & Szepesvári, 2008; Bagnell et al., 2003; Rajeswaran et al., 2017). Strategies such as $\epsilon$-greedy or Gaussian exploration have recently been used to successfully solve a large number of dense reward problems. As the reward becomes sparse or heavily delayed, such strategies be-

come intractable in high-dimensional settings. Critically, these approaches perform exploration on a per time-step basis, which can lead to back and forth wandering preventing efficient exploration. Parameter-space exploration (Plappert et al., 2017; Fortunato et al., 2017) methods do not explore at each time step, but rather generate correlated behaviors based on explored parameters at the start. However, such approaches do not consider exploration as an intentional act, but is rather a deviation from a well defined objective for the agent. Deep exploration strategies (Osband et al., 2013) sample a value function from the posterior and use it for greedy action selection. Approaches based on notions of intrinsic motivation and information gain (Chentanez et al., 2005; Stadie et al., 2015; Houthooft et al., 2016; Pathak et al., 2017; Bellemare et al., 2016) also explicitly introduce exploration bonuses into the agent's reward system. However, such approaches critically do not have the element of planning to explore; thus the agent may not actually reach regions of high predicted reward because it does not know how to get there. Our work is perhaps closest to the $E3$ framework of Kearns & Singh (2002), which considers altered MDPs with different reward functions, and executes the optimal action under that MDP. However solving these altered MDPs is expensive and their solution is quickly discarded. MPC on the other hand can quickly solve for local instance-specific solutions in these MDPs.

**Model-free RL:** Our work investigates how much training times can be reduced over model-free methods when the internal model is an accurate representation of the world model. As a representative number, Schulman et al. (2015) report approximately 5 days of agent experience and 128 CPU core hours for solving tasks such as getting up from the ground. In contrast, POLO requires only 12 CPU core hours and 96 seconds of agent experience. Recently, policy gradient methods were also used for in-hand manipulation tasks (OpenAI, 2018), where 3 years of simulated experience and 500 CPU hours were used for object reorientation tasks. For a similar task, POLO only required 1 CPU hour. Of course, model-free methods do not require an accurate internal model, but our results suggest that much less experience may be required for the control aspect of the problem. Our work can be viewed as a strong model-based baseline that model-free RL can strive to compete with, as well as a directly useful method for researchers studying simulation to reality transfer.

In an alternate line of work, internal models have been used for variance reduction purposes in model-free RL (Feinberg et al., 2018; Buckman et al., 2018), in contrast to our use of MPC. Related to this, Azizzadenesheli et al. (2018b) consider learning an internal model for discrete action domains like ALE and use short horizon MCTS for planning. Similarly, Nagabandi et al. (2018) learn a dynamics model in simple continuous control tasks and use a random shooting MPC method for action selection. These lines of work consider the interplay between learning dynamics models and planning procedures, and try to improve the quality of internal models. As a consequence, they focus on domains where simple action selection procedures with accurate models obtain near-optimal performance. In our work, we show that we can learn value functions to help real-time action selection with MPC on some of the most high-dimensional continuous control tasks studied recently. Thus, the two lines of work are complementary, and combining POLO with model learning would make for an interesting line of future work.

## 5 Conclusions and Future Work

In this work we presented POLO, which combines the strengths of trajectory optimization and value function learning. In addition, we studied the benefits of planning for exploration in settings where we track uncertainties in the value function. Together, these components enabled control of complex agents like 3D humanoid and five-fingered hand. In this work, we assumed access to an accurate internal dynamics model. A natural next step is to study the influence of approximation errors in the internal model and improving it over time using the real world interaction data.

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

# A    Appendix: Experimental Details, Humanoid

The model used for the humanoid experiments was originally distributed with the MuJoCo (Todorov et al., 2012) software package and modified for our use. The model nominally has 27 degrees of freedom, including the floating base. It utilizes direct torque actuation for control, necessitating a small timestep of 0.008 seconds. The actuation input is limited to $\pm 1.0$, but the original gear ratios are left unchanged.

For POLO, the choice of inputs for the value function involves a few design decisions. We take inspiration from robotics by using only easily observed values.

| Dims. | Observation |
|---|---|
| 6 | Direction & Normal Vector, Torso |
| 3 | Direction Vector, Neck to R. Hand |
| 3 | Direction Vector, Neck to L. Hand |
| 3 | Direction Vector, Hip to R. Foot |
| 3 | Direction Vector, Hip to L. Foot |
| 5 | Height, Root, Hands, & Feet |
| 6 | Root Velocities |
| 5 | Touch Sensors, Head, Hands, & Feet |

| Value | Parameter |
|---|---|
| 0.99 | $\gamma$ discount Factor |
| 64 | Planning Horizon Length |
| 120 | MPPI Rollouts |
| 0.2 | MPPI Noise $\sigma$ |
| 1.25 | MPPI Temperature |

For value function approximation in POLO for the humanoid tasks, we use an ensemble of 6 neural networks, each of which has 2 layers with 16 hidden parameters each; *tanh* is used for non-linearity. Training is performed with 64 gradient steps on minibatches of size 32, using ADAM with default parameters, every 16 timesteps the agent experiences.

In scenarios where the agent resets, we consider a horizon of 600 timesteps with 20 episodes, giving a total agent lifetime of 12000 timesteps or 96 seconds. When we consider no resets, we use the same total timesteps. A control cost is shared for each scenario, where we penalize an actuator's applied force scaled by the inverse of the mass matrix. Task specific rewards are as follows.

## A.1    Humanoid Getup

In the getup scenario, the agent is initialized in a supine position, and is required to bring its root height to a target of 1.1 meters. The reward functions used are as follows. In the non-sparse case, the difficulty in this task is eschewing the immediate reward for sitting in favor of the delayed reward of standing; this sequence is non-trivial to discover.

$$R(s) = \begin{cases} 1.0 - (1.25 - Root_z), & \text{if } Root_z \leq 1.25 \\ 1.0, & \text{otherwise} \end{cases}, R_{sparse}(s) = \begin{cases} 0.0, & \text{if } Root_z \leq 1.25 \\ 1.0, & \text{otherwise} \end{cases}$$

## A.2    Humanoid Walk

In the walking scenario, the agent is initialized in an upright configuration. We specify a reward function that either penalizes deviation from a target height of 1.1 meters, or penalizes the deviation from both a target speed of 1.0 meters/second and the distance from the world's x-axis to encourage the agent to walk in a straight line. We choose a target speed as opposed to rewarding maximum speed to encourage stable walking gaits.

$$R(s) = \begin{cases} -(1.25 - Root_z), & \text{if } Root_z \leq 1.25 \\ 1.0 - |1.0 - Vel_x| - |Root_x|, & \text{otherwise} \end{cases}$$

## A.3    Humanoid Box

For the box environment, we place a 0.9 meter wide cube in front of the humanoid, which needs to be pushed to a specific point. The friction between the box and ground is very low, however, and most pushes cause the box to slide out of reach; POLO learns to better limit the initial push to

control the box to the target.

$$R(s) = \begin{cases} -(1.25 - Root_z), & \text{if } Root_z \leq 1.25 \\ 2.0 - ||Box_{xy} - Root_{xy}||_2, & \text{else if } |Box_{xy} - Root_{xy}|_2 > 0.8 \\ 4.0 - 2 * ||Box_{xy} - Target_{xy}||_2, & \text{otherwise} \end{cases}$$

In this setup, the observation vector increases with the global position of the box, and the dimensionality of the system increase by 6. The box initially starts 1.5 meters in front of the humanoid, and needs to be navigated to a position 2.5 meters in front of the humanoid.

## B  APPENDIX: EXPERIMENTAL DETAILS, HAND MANIPULATION

We use the Adroit hand model (Kumar, 2016) and build on top of the hand manipulation task suite of Rajeswaran et al. (2018). The hand is position controlled and the dice is modeled as a free object with 3 translational degrees of freedom and a ball joint for three rotational degrees of freedom. The base of the hand is not actuated, and the agent controls only the fingers and wrist. The dice is presented to the hand initially in some randomized configuration, and the agent has to reorient the dice to the desired configuration. The desired configuration is randomized every 75 timesteps and the trajectory optimizer does not see this randomization. Thus the randomization can be interpreted as unmodelled external disturbances to the system. We use a simple reward function for the task:

$$R(s) = -0.5 \, \ell_1(x_o, x_g) - 0.05 \, \ell_{quat}(q_o, q_g),$$

where $x_o$ and $x_g$ are the Cartesian positions of the object (dice) and goal respectively. The goal location for the dice is a fixed position in space and is based on the initial location of the palm of the hand. $\ell_1$ is the L1 norm. $q_o$ and $q_g$ are the orientation configurations of object and goal, respectively, and expressed as quaternions with $\ell_{quat}$ being the quaternion difference.

We use 80 trajectories in MPPI with temperature of 10. We use an ensemble of 6 networks with 2 layers and 64 units each. The value function is updated every 25 steps of interaction with the world, and we take 16 gradient steps each with a batch size of 16. These numbers were arrived at after a coarse hyperparameter search, and we expect that better hyperparameter settings could exist.

## C  PROOF OF LEMMA 2 AND REMARKS

Let $\hat{\tau}$ and $\tau^*$ represent the trajectories of length $H$ that would be generated by applying $\hat{\pi}_{MPC}$ and $\pi^*$ respectively on the MDP. Starting from some state $s$, we have:

$$V^*(s) - V^{\hat{\pi}_{MPC}}(s) = \mathbb{E}_{\tau^*}\left[\sum_{t=0}^{H-1} \gamma^t r_t + \gamma^H V^*(s_H)\right] - \mathbb{E}_{\hat{\tau}}\left[\sum_{t=0}^{H-1} \gamma^t r_t + \gamma^H V^{\hat{\pi}_{MPC}}(s_H)\right] \quad (9)$$

Adding and subtracting, $\mathbb{E}_{\hat{\tau}}[\sum_t \gamma^t r_t + \gamma^H V^*(s_H)]$, we have:

$$V^*(s) - V^{\hat{\pi}_{MPC}}(s) = \gamma^H \mathbb{E}_{\hat{\tau}}\left[V^*(s_H) - V^{\hat{\pi}_{MPC}}(s_H)\right]$$
$$+ \mathbb{E}_{\tau^*}\left[\sum_{t=0}^{H-1} \gamma^t r_t + \gamma^H V^*(s_H)\right] - \mathbb{E}_{\hat{\tau}}\left[\sum_{t=0}^{H-1} \gamma^t r_t + \gamma^H V^*(s_H)\right]. \quad (10)$$

Since $\max_s |\hat{V}(s) - V^*(s)| = \epsilon$, we have:

$$\mathbb{E}_{\tau^*}\left[\sum_{t=0}^{H-1} \gamma^t r_t + \gamma^H V^*(s_H)\right] \leq \mathbb{E}_{\tau^*}\left[\sum_{t=0}^{H-1} \gamma^t r_t + \gamma^H \hat{V}(s_H)\right] + \gamma^H \epsilon \quad (11)$$

$$\mathbb{E}_{\hat{\tau}}\left[\sum_{t=0}^{H-1} \gamma^t r_t + \gamma^H V^*(s_H)\right] \geq \mathbb{E}_{\hat{\tau}}\left[\sum_{t=0}^{H-1} \gamma^t r_t + \gamma^H \hat{V}(s_H)\right] - \gamma^H \epsilon \quad (12)$$

Furthermore, since $\hat{\tau}$ was generated by applying $\hat{\pi}_{MPC}$ which optimizes the actions using $\hat{V}$ as the terminal value/reward function, we have:

$$\mathbb{E}_{\hat{\tau}}\left[\sum_{t=0}^{H-1} \gamma^t r_t + \gamma^H \hat{V}(s_H)\right] \geq \mathbb{E}_{\tau^*}\left[\sum_{t=0}^{H-1} \gamma^t r_t + \gamma^H \hat{V}(s_H)\right] \quad (13)$$

using these bounds, we have:

$$
\begin{aligned}
V^*(s) - V^{\hat{\pi}_{MPC}}(s) &\leq \gamma^H \mathbb{E}_{\hat{\tau}}\left[V^*(s_H) - V^{\hat{\pi}_{MPC}}(s_H)\right] + 2\gamma^H \epsilon \\
&\leq 2\gamma^H \epsilon \left(1 + \gamma^H + \gamma^2 H + \ldots\right) \\
&\leq \frac{2\gamma^H \epsilon}{1 - \gamma^H}
\end{aligned}
\tag{14}
$$

by recursively applying the first bound to $V^*(s_H) - V^{\hat{\pi}_{MPC}}(s_H)$. This holds for all states, and hence for any distribution over states.

**Notes and Remarks:** For Eq. (13) to hold in general, and hence for the overall bound to hold, we require that the actions are optimized in closed loop. In other words, MPC has to optimize over the space of feedback policies as opposed to open loop actions. Many commonly used MPC algorithms like DDP and iLQG Jacobson & Mayne (1970); Todorov & Li (2005) have this property through the certainty equivalence principle for the case of Gaussian noise. For deterministic dynamics, which is the case for most common simulators like MuJoCo, Eq. (13) holds without the closed loop requirement. We summarize the different cases and potential ways to perform MPC below:

- In the case of deterministic dynamics, the optimal open loop trajectory and optimal local feedback policies have the same performance up to finite horizon $H$. Thus, any trajectory optimization algorithm, such as iLQG and MPPI can be used.

- In the case of stochastic dynamics with additive Gaussian noise, local dynamic programming methods like iLQG and DDP Todorov & Li (2005); Jacobson & Mayne (1970) provide efficient ways to optimize trajectories. These approaches also provide local feedback policies around the trajectories which are optimal due to the certainty equivalence principle.

- In the case of general stochastic systems, various stochastic optimal control algorithms like path integral control Theodorou et al. (2010) can be used for the optimization. These situations are extremely rare in robotic control.

Finally, we also note that Sun et al. Sun et al. (2018a) propose and arrive at a similar bound in the context of imitation learning and reward shaping. They however assume that a policy can simultaneously optimize the approximate value function over $H$ steps, which may not be possible for a parametric policy class. Since we consider MPC which is a non-parametric method (in the global sense), MPC can indeed simultaneously optimize for $H$ steps using $\hat{V}$.

