# OpenReview forum: "Plan Online, Learn Offline: Efficient Learning and Exploration via Model-Based Control"
_ICLR.cc/2019/Conference_

### Official Review · AnonReviewer1 · 2018-10-31
**Nice results but lean technical contribution**

**Rating:** 4
**Confidence:** 3

**Review:**

This paper proposes to combine fitted value iteration with model predictive control (MPC) to speed up the learning process. The value iteration is the "Learn offline" subsystem while MPC is the "Plan online" subsystem. In addition, this paper also proposes an exploration technique that increases exploration if the multiple value function estimators disagree. The evaluation is complete and shows nice results.

However, I did not rank this paper high for two reasons. First, it is not clear to me how the model is acquired in MPC. Does the method learn the model? Does the method linearize the dynamics and assume a linear model? I am not sure. I suspect that the method just uses the simulator as the model. If it is the case, the method is not so useful because for complexity systems, such as humanoids, we do not know the model. And the comparisons with model-free learning algorithms are not fair because the paper assumes that the model is given. If this is not the case, I suggest that a more detailed description of MPC should be presented in Section 2.3.

Second, the technical contributions are lean. The three main components, 1) fitted value iteration, 2) MPC and 3) exploration based on multiple value function estimates, are not novel. The combination of them seems straight forward. For example, the H-step Bellman update (Section 2.3) is a blend between Monte-Carlo method and Q learning. It seems to be similar to the TD(\lambda) method. Thus, it is not surprising that it can accelerate convergence of value function.

For the above reasons, I would not recommend accepting this paper at this time.

---

> ### Author Response · Authors · 2018-11-15
> **Response to review**
>
> Thank you for taking time to review our paper, and for your analysis and review. We address your two concerns as follows:
>
> =======================
> Regarding source of model
> =======================
> In this work, we assume that we have access to the dynamics model of the environment. We do not believe that this is a severe limitation because reasonable dynamics models are known for a majority of complex engineered systems including robotics. Indeed, most success stories in robotics are through use of models/simulators. Notable examples include Boston Dynamics’ Atlas, Honda’s Asimo, and the recent in-hand manipulation results from OpenAI. There is also a growing body of work and interest in simulation to reality transfer in RL for robotics, and we believe that POLO would serve as a strong baseline method for this research direction. POLO is also complementary to fields like learning dynamics models and nonlinear system identification.
>
> Furthermore, we also want to emphasize that knowing the dynamics does not make the problem trivial. Certain aspects of the MDP may be revealed only at run-time, thereby ruling out the option of essentially pre-solving the MDP before deployment time. For instance, we may not know the states to concentrate the approximation power of policy search or dynamic programming methods till deployment time. The reward function may also be revealed only at deployment time (robot knows physics, but does not know what task to do till the human tells it). Thus, having algorithms that can compute good actions at run-time is critical for a variety of settings, and we show in our results that POLO outperforms MPC.
>
> =======================
> Comparisons to model-free RL
> =======================
> Model-free RL does not assume explicit knowledge of the dynamics, which is certainly a weaker assumption that in the POLO case. However, model-free RL has predominantly been demonstrated only in simulated environments where a model is available by definition (e.g. AlphaGo, Atari, MuJoCo). We believe that POLO would be an important contribution for researchers studying simulation to reality transfer, since it is orders of magnitude more efficient than running model-free RL in simulators. We have updated the paper to reflect this comparison more accurately.
>
> =======================
> Significance and novelty
> =======================
> POLO combines three important and deep threads of work: MPC, approximate dynamic programming, and exploration. The primary contribution of this work is to connect the three threads of work, into a simple and elegant algorithm, as opposed to making a contribution to any one of the streams. We believe that combining the threads of work into a practical algorithm that produces impressive results to be important and valuable. We emphasize that all reviewers found the motivation and presentation of the algorithmic framework to be elegant. While the combination of MPC and value function may appear seemingly straightforward, it has not been found effective in continuous control in the past. For example, Zhong et al. study the setting of learning value function to help MPC and found the contribution of the value function to be minimal in their settings. We also emphasize that combining MPC and uncertainty quantification to do efficient and targeted exploration for continuous control has not been studied in the past.
>
> =======================
> Summary
> =======================
> To summarize, while each component of POLO is well studied, combining them into a practical algorithm that produces impressive results is far from obvious. Combining the different components allows POLO to synthesize and learn competent behaviors on the fly for high dimensional systems. While we assume to know the dynamics model, this is not an outlandish assumption given the prevalence of complex model based robotic control in the real world, and the growing body of work in learning dynamics models, intuitive physics, and simulation to reality transfer.

---

### Official Review · AnonReviewer2 · 2018-11-02
**limited insight and novelty**

**Rating:** 5
**Confidence:** 5

**Review:**

In this paper, the authors propose POLO, a reinforcement learning algorithm which has access to the model of the environment and performs RL to mitigate the planning cost. For the planning, POLO uses the known model of the environment up to a fixed horizon H and then use an approximated value function in the leaf nodes. This way, instead of planning for an infinite horizon, the planning is factored to a shorter horizon, resulting in lower computation cost.

The novelty and motivation behind this approach is limited. Similar or even more general approach for discrete action space is introduced in "Sample-Efficient Deep RL with Generative Adversarial Tree Search" where they also learn the model of the environment and additionally consider the error due to the model estimation. There is also a clear motivation in the mentioned paper while I could not find a convincing one for the current paper.
Putting the novel limitation aside,  both of these paper, the current paper, and the paper I mentioned, suffer from very lose estimation bounds. Both of these works bound somewhat similar (not the same) things via L_inf error of value function which in practice does not necessarily result in useful or insightful upper bounds (distribution dependent bound is desired). Moreover, with the assumption of knowing the environment model, the implication of the current work is significantly limited.

The authors do a good job of writing the paper and the paper is clear which is appreciatable.

In equation 6 the authors use log-sum-exp and claim it corresponds to UCB, but they do not provide any evidence to support their claim.

In addition, the Bayesian linear regression in the tabular setting is firstly proposed in Generalization and Exploration via Randomized Value Functions and beyond tabular setting (the setting in the current paper) was proposed in Efficient Exploration through Bayesian Deep Q-Networks.

The claims in this paper are not strong enough and the empirical study does not strongly support or provide sufficient insight. For example experiments in section 3.2 does not provide much insight beyond common knowledge.

While bridging the gap between model based and model free approaches in RL are significantly important research directions in RL, I do not find the current draft significant enough to shed sufficient light into this topic.

---

> ### Author Response · Authors · 2018-11-15
> **Response to review (1/2)**
>
> Thank you for taking time to review our paper and for the feedback. We address your concerns below, and hope that our clarifications would help appreciate the work better. We look forward to continued fruitful discussions.
>
> =======================
> Significance & Novelty
> =======================
> POLO combines three important and deep threads of work: MPC, approximate dynamic programming, and exploration. The primary contribution of this work is to connect the three threads of work as opposed to making a contribution to any one. We believe that combining them into a simple and elegant algorithm that produces impressive results is important and valuable. We emphasize that all reviewers found the motivation and presentation of the algorithmic framework to be elegant. While the combination of MPC and value function may seem straightforward, it has not found wide applicability in continuous control settings in the past. For example, Zhong et al. study the setting of learning value function to help MPC and found the contribution of the value function to be minimal in their settings. We also emphasize that combining MPC and uncertainty quantification to do efficient and targeted exploration has not been studied in the past in continuous control settings.
>
> Our empirical study attempts to isolate individual benefits enabled by each component in the POLO framework. Firstly, we have clearly demonstrated that learned value functions can support short horizon MPC. This has not been explored extensively in controls applications, and most MPC works do not consider learning a value function using the interaction data. Secondly, we demonstrate the utility of uncertainty quantification and MPC for exploration, through the maze example. Further, we demonstrate that MPC accelerates value function learning. While individual components may have been suggested before (Bellman himself suggests using prior experience to reduce planning computation), we present all the benefits in one elegant framework that actually achieves very strong empirical results in practice as noted by other reviewers.
>
> =======================
> Known dynamics model
> =======================
> First and foremost, we emphasize that in the known dynamics setting our algorithm significantly outperforms model-free RL methods like policy gradient. While model-free RL obviously does not require access to a model, the overwhelming majority of results in RL are in simulated environments (e.g. AlphaGo, Atari etc.) where a model is available by design. Furthermore, the majority of successful results in robotics are also through model-based methods (eg Boston Dynamics' Atlas, Honda's Asimo, OpenAI's dexterous hands). Thus, one can interpret POLO as a very strong model-based baseline that model-free RL algorithms can strive to compete with, or as a powerful vehicle with direct applicability for simulation to reality transfer, which is a topic of immense interest in robot learning.
>
> Furthermore, we wish to point out that knowing the dynamics does not make the problem trivial. Certain aspects of the MDP may be revealed only at run-time, thereby ruling out the possibility of pre-solving the MDP. For instance, we may not know the states to concentrate the approximation power of policy search or dynamic programming methods till deployment time. The reward function may also be revealed only at deployment time (robot knows physics, but does not know what task to do till the human tells it). Thus, having algorithms that can compute good actions at run-time is critical for a variety of settings, and we show in our results that POLO outperforms MPC.
>
> Finally, we wish to point out that we make explicit the assumption of knowing the dynamics model, and do not even attempt to bridge model-free and model-based RL methods (as used in the connotation of recent papers). We feel that it is important to not judge the work on the basis of a problem we are not attempting to solve.

---

> > ### Author Response · Authors · 2018-11-15
> > **Response to review (2/2)**
> >
> > =======================
> > Related Works
> > =======================
> > Thank you for pointing out the GATS paper, we have included a citation to it in our updated submission. As discussed earlier, the broad idea of combining planning and value function learning is not new. However, intuitions and lessons learned from discrete settings rarely transfer to continuous domains. For instance, global value or Q learning methods have not produced great results in continuous control with high-dimensional action spaces, while DQN performs very well in Atari which has a small number of discrete actions. Similarly, very different planning approaches are used in discrete action settings (e.g. UC-Trees) and continuous robotics problems (e.g. iLQG, PI^2, RRT). We emphasize that in the continuous control settings, we can synthesize controllers orders of magnitude more efficient than currently used approaches like PPO in the OpenAI dexterous hand work.
> >
> > Thanks for pointing out the other papers studying Bayesian linear regression, we have included citations to those as well. We would like to emphasize that the computational view of Bayesian regression is not the contribution of this work. Rather, we use it as a means to perform uncertainty estimation and drive exploration in the POLO framework.
> >
> > =======================
> > Answers to other questions
> > =======================
> > - Regarding equation 6, we actually *do not* claim that our approach corresponds to UCB. Rather, we only say that log-sum-exp is a risk seeking objective and corresponds to optimism in the face of uncertainty, and this broad heuristic has been used successfully in other works.
> > - Regarding Lemma 2, this is not a primary contribution of our work, and is fairly elementary. We use it primarily as a motivation for the practical algorithm we develop. We agree that the L_inf norm bounds are loose and tighter bounds would be great, but that is orthogonal to the main points of this paper.
> >
> > =======================
> > Summary
> > =======================
> > In summary, we have presented an elegant framework and algorithm that offers tangible benefits in the space of continuous control. This enables solutions to complex control problems orders of magnitude more efficiently than currently used techniques. The work should be evaluated based on the clean presentation and strong empirical results as opposed to weak connections to problems and bounds we do not focus on.

---

### Official Review · AnonReviewer3 · 2018-11-03
**Lucid paper with nice ideas, but problem setting not completely clear**

**Rating:** 6
**Confidence:** 4

**Review:**

This paper was a joy to read.   The description and motivation of the POLO framework was clear, smart, and sensible.  The fundamental idea is to explore the interplay between value-function estimation and model-predictive control and demonstrate how they benefit one another.  None of these ideas is fundamentally new, but the descriptions and their combination is very nice.

As I finished the paper, though, I was left with a lingering lack of understanding of the exact problem setting that is being addressed. The name is cute but didn't help clarify.    As I understand it:
- we have a correct dynamics model (I'm assuming that's what "nominal dynamics model" means) and a good trajectory optimization algorithm
- the agent has limited online cognitive capacity
- there is no opportunity for offline computation
If offline computation time were available, then we could run this algorithm (or your favorite other RL algorithm) in the agent's head before taking any actions in the actual world.   That does not seem to be the setting here, although it does seem to me that you might be able to show that POLO is a good algorithm for finding a value function, offline, with no actual interaction with the world.

So, fundamentally, this paper is about action under computational time constraints.   One strategy would be for the robot to use 7 of its cores to run your favorite approximate DP / RL algorithm in parallel with 1 core that's used for action selection.  Why is that worse than your algorithm 1?

Setting this question aside, I had some other comments:
- It is better *not* to use "trajectory optimization" and "model-predictive control" interchangeably.  I can use traj opt in other circumstances (e.g. with open loop trajectory following) and could use other planners for MPC.
- Some version of lemma 2 probably (almost certainly) already exists somewhere in the literature;  I'm sorry, though, that I can't point you to a concrete reference.
- The argument about MPC letting us approximate H Bellman backups is plausible, but seems somewhat subtle;  it would be good to elaborate it in some more detail.
- The set of assertions and experiments is very nice.
- Why are no variances shown in figure 3?   Why does performance seem to degrade after a certain horizon.

This paper doesn't seem really to be about learning representations.  I don't know if that's important to the ICLR decision-making.

---

> ### Author Response · Authors · 2018-11-15
> **Response to review**
>
> Thank you for taking time to review our paper and for the constructive feedback. We greatly appreciate the comment that you enjoyed the exposition, assertions, and results in the paper. We look forward to continued fruitful discussions!
>
> ==================
> Problem Setting
> ==================
> The agent knows the MDP dynamics, but the MDP can be very complex with some information about the MDP revealed only at deployment time. Hence, it is not feasible in general to “pre-solve” the MDP and simply deploy the solution. For instance, we may know the state distribution only at deployment time and hence not know where to concentrate the approximation power in policy gradient or dynamic programming methods. Also, the reward function may be revealed only at deployment time (the robot knows physics but doesn’t know which task to do until human command). This is the general premise of real-time MPC which has enjoyed tremendous success in controlling complex systems in engineering and robotics. At the same time, we note that if there is a possibility to pre-solve the MDP before deployment, POLO can be used for this purpose as well and our experiments show that POLO is more efficient than fitted value iteration.
>
> =======================
> Significance and novelty
> =======================
> First and foremost, we emphasize that POLO produces very impressive results for hard continuous control tasks as noted by all the reviewers. POLO requires 1 CPU hour as opposed to 500 CPU hours reported by OpenAI (our numbers with PG are similar as well, and we will include these with the final paper). While model-free RL obviously does not require access to a model, the overwhelming majority of results in RL (e.g. AlphaGo, Atari, MuJoCo) are in simulated environments where a model is available by design. Model based methods have also been very successful in robotics (e.g. Boston Dynamics’ Atlas, Honda’s Asimo, OpenAI’s dexterous hands). Thus, we believe that knowing the dynamics model is not a severe limitation. One can interpret POLO as a very strong model-based baseline that model-free RL algorithms can strive to compete with, or as a powerful vehicle with direct applicability for simulation to reality transfer, which is a topic of immense interest in robot learning.
>
> POLO combines three important and deep threads of work: MPC, approximate dynamic programming, and exploration. The primary contribution of this work is to connect the three threads as opposed to making a contribution to any one. We believe that combining these threads of work into a simple and elegant algorithm that produces impressive results to be important and valuable. We emphasize that all reviewers found the motivation and presentation of the algorithmic framework to be elegant. Furthermore, combining MPC and uncertainty quantification to do efficient and targeted exploration has not been explored in the past in continuous control.
>
> =======================
> Reg. alternate approach
> =======================
> You are indeed correct that the core question is about action selection with bounded resources at run-time. In this setting, using any RL/DP algorithm on 7 cores, it is natural to focus the search process around the current state of interest due to limited resources. Thus, the suggested approach reduces to MPC -- 7 cores perform local rollouts which are then combined by the final core in some way -- either non-parametric blending with exponentiated costs (MPPI), a fitted form of iLQG, or some alternative. We show in our results that POLO outperforms trajectory centric RL which is synonymous with MPC.
>
> =======================
> Additional comments
> =======================
> We will include additional discussion about the following suggested components in the final version: (a) trajectory optimization vs MPC; (b) H-step Bellman backups; (c) error bars for the plots.
> We agree that trajectory optimization has broader connotations than MPC. In this work, we used it in the context of real-time trajectory optimization which is synonymous with MPC. We will clarify the distinctions in the paper.
> We also emphasize that Lemma 2 is not a primary contribution of the paper -- it primarily serves as a motivation for the algorithm we develop. We agree that prior work should have Lemma 2, since it is fairly elementary, and will include additional citations if we find the appropriate sources.

---

### Author Response · Authors · 2018-11-17
**Summary of responses**

We thank all the reviewers and the area chair for taking time to read our paper and providing feedback. The summary of our responses to common questions raised by the reviewers is below. We look forward to continued discussions to address any additional questions.

>>> Source of dynamics model

In this work, we assume that we have access to the ground truth dynamics model. We *do not* believe that this is an unreasonable assumption, especially for our motivating problems. Good models based on knowledge of physics or through learning (system identification) are available in most engineering and robotics settings. Indeed, most successful robotics results have been through use of models or simulators (Boston Dynamics, Honda, OpenAI). The work is also directly relevant for fields where dynamics models are available (e.g. character animation in graphics) and simulation to reality transfer, which is gaining a lot of interest in robotic learning.

We also emphasize that knowing the dynamics does not make the problem trivial, and does not imply that one can simply “pre-solve” the MDP and deploy the solution. Some aspects of the MDP may be revealed only at deployment time, such as the states where we may want to concentrate the approximation power, or the reward function may be revealed only at deployment time (robot knows physics but does not know which task to solve). We thus feel that algorithms for real-time action selection is an important component to enable robots to behave competently in dynamic environments.

>>> Novelty

POLO combines three threads of work into one coherent and elegant algorithm that produces impressive results. All reviewers have pointed out and noted that the motivation and presentation of the algorithm is clear and neat, and the results are impressive. While it may be easy to postulate that bringing together these threads of work is important, the specifics of how to do this to produce robust algorithms with impressive results is highly non-trivial and far from obvious. We feel that the quality of results should be taken into account when assessing the novelty. Indeed, one could argue that landmark results like AlphaGo and AlphaZero do not make deep contributions to any sub-field of RL/ADP, but it remains one of the most impressive feats due to bringing together different algorithmic sub-components and showing impressive results. We also note that the component of planning/MPC to explore, which we demonstrate in the maze example, has not been explored in continuous control.


We hope that the above clarifications also help to resolve other questions that were raised. In particular, our goal is *not* to bridge model-free and model-based RL methods; nor is to provide strong performance bounds for MPC. We make no claims about the former, and we merely use the latter as a motivation to develop a practical algorithm. While these are very important questions, they are not our focus and beyond the scope of the current submission. We kindly request that our paper be evaluated on the basis of results for the problem setting we study, as opposed to insights to other problem domains/settings.

---

### Comment · Area_Chair1 · 2018-11-20
**additional reviewer input, after consideration of author responses ?**

Thank you to everyone for the detailed reviews and the authors for their detailed responses.

Now would be a great time to hear from the reviewers as to whether their concerns
have been addressed, and if they wish to make any score adjustments.

Thanks in advance for this additional input.
-- area chair

---

### Author Response · Authors · 2018-12-15
**Any additional questions from the reviewers?**

Dear Reviewers,

Thank you once again for taking time to review our paper. Please let us know if we can answer any additional questions about the work, or if the answers to any of your questions require further discussion. If the responses were satisfactory, we kindly request that the reviewers adjust the rating appropriately. Thank you once again and we look forward to additional discussions about the work!

---

### Meta-Review · Area_Chair1 · 2018-12-16
**Useful combination of MPC, approximate-DP, and value-function ensembles**

**Confidence:** 4
**Recommendation:** Accept (Poster)

**Metareview:**

The paper makes novel explorations into how MPC and approximate-DP / value-function approaches, with value-fn ensembles to model value-fn uncertainty, can be effectively combined.  The novelty lies in exploring their combination. The experiments are solid. The paper is clearly written.

Open issues include overall novelty, and delineating the setting in which this method is appropriate.

The reviewers and AC are in agreement on what is in the paper. The open question is whether
the combination of the ideas is interesting.

After further reviewing the paper and results. the AC believes that the overall combination of ideas and related evaluations that make a useful and promising contribution. As evidenced in some of the reviewer discussion, there is often a
considerable schism in the community regarding what is considered fair to introduce in terms of
prior knowledge, and blurred definitions regarding planning and control. The AC discounted some of the
concerns of R2 that related more to discrete action settings and theoretical considerations; these
often fail to translate to difficult problems in continuous action settings.  The
AC believes that R3 nicely articulates the issues of the paper that can be (and should be) addressed in the writing, i.e., to
describe and motivate the settings that the proposed framework targets, as articulated in the reviews and ensuing discussion.